# The Role of Job Resources in the Relationship between Job Demands and Work-Related Musculoskeletal Disorders among Hospital Nurses in Thua Thien Hue Province, Vietnam

**DOI:** 10.3390/ijerph19084774

**Published:** 2022-04-14

**Authors:** Hai Ba Mai, Jiyun Kim

**Affiliations:** 1Faculty of Nursing, Hue University of Medicine and Pharmacy, Hue University, Hue 47000, Vietnam; mbhai@huemed-univ.edu.vn or; 2School of Nursing, Gachon University, Incheon 21936, Korea

**Keywords:** job stress, workload, social support, nurses

## Abstract

(1) Background: This cross-sectional study aimed to determine the moderating effect of job resources (JR) in the relationship between job demands (JD) and work-related musculoskeletal disorders (WRMSDs) among hospital nurses in the Thua Thien Hue province, Vietnam. (2) Methods: Data were collected via a self-reporting questionnaire administrated to 225 nurses from two hospitals in the Thua Thien Hue province of Vietnam from August to September 2020. The questionnaire included information on musculoskeletal symptoms, JD and JR, and the demographics and job characteristics of participants. The simple moderation analysis was used for data analysis. (3) Results: The findings indicated that 87.6% of nurses had experienced WRMSDs in at least one body region during the previous 12 months, with lower back (65.3%), neck (61.8%), and shoulders (61.8%) being the three most common sites. Physical workloads (PW) and psychological demands (PDs) were significantly correlated with WRMSDs severity (*p* < 0.05). This study further underlined the moderating role of coworker support (CS) on the relationship between PW and WRMSD severity. WRMSDs severity significantly increased when PW was high alongside a low level of CS. (4) Conclusions: The current study indicated that nurses should be provided with appropriate CS to minimize WRMSDs severity. Furthermore, an intervention program for managing WRMSDs in nurses may involve minimizing physical risk factors and PDs.

## 1. Introduction

Work-related musculoskeletal disorders (WRMSDs) are reportedly the most common occupational health problem in the healthcare sector [1], with the prevalence of WRMSDs worldwide being higher in nurses than any other work-related group [2,3]. Neck pain, shoulder pain, and lower back pain have been the most reported symptoms [3,4]. These symptoms induce functional activity reduction and decrease job performance and working productivity [5]. WRMSDs can also result in workers needing a leave of absence from work [6], becoming unable to continue work [7,8], and presenteeism [9,10]. Furthermore, these effects worsen the health outcomes of nurses and increase healthcare costs [11].

Previous studies have indicated that the occurrence of WRMSDs is frequently complex and affected by individual characteristics and physical and psychosocial risk factors [12,13,14]. The relationship between these risk factors and health problems has also previously been tested using various models, such as the US National Research Council’s model of musculoskeletal disorders (MSD) hazards and risk factors (2001), and the demands-control model [15], which was later called the job–demand–control–support model [16]. However, most studies of these models have been restricted to a very limited sets of independent variables that may not be applicable to all job and worker types [17]. According to the job demands–resources (JD-R) model [17,18], each occupation is assumed to have distinct risk factors for work-related health problems. These risk factors are divided into the two categories of job demands (JD) and job resources (JR) [17]. JD refer to the physical, psychological, and social aspects of jobs that require sustained physical and/or psychological effort or skills for employees to perform their work, and are therefore associated with health issues [19]. JR refer to the physical, psychological, social, or organizational aspects of work that function to achieve job objectives, and lower JD, physiological and psychological costs [17,18]. This model suggests that health problems are more likely to arise when a high JD and lack of JR are combined in a working environment. This suggests that JR can help to alleviate job strain by providing employees with methods to meet their work demands [19,20], or by preventing them from harm and health problems [17,21]. The JD-R model was therefore used as research framework for the present study.

Recent studies have suggested that the interaction between JD and decision latitude or social support can also predict strain [22], and also the perception of work-related health [23]. Skovlund et al. (2020) suggested that high physical workloads (PWs) combined with high-strain conditions are associated with an increased risk of MSDs [24]. This indicates that MSDs in the workplace might be altered by a combination of physical and psychosocial demands and JR. Most previous studies have indicated these risk factors independently influence the prevalence and incidence of WRMSDs [7], but there is little evidence for the hypothesized association between JD and JR in predicting WRMSDs. Furthermore, most previous studies examined these relationships using symptom presence or the anatomical sites of MSDs [25,26,27,28]. Research on the severity of WRMSD symptoms is scarce. Analyzing the factors that moderate the relationship between JD and WRMSD severity is an important strategy for understanding symptom characteristics and their risk factors, and to develop a program to more effectively reduce symptom severity and negative workplace outcomes.

Nurses and midwives constitute the majority of the workforce in the Vietnamese healthcare system. It is estimated that 140,000 nurses and midwives account for more than half of the healthcare workforce, who provide the most frequent and continuous services, and thereby serve as the backbone of the health sector. A report by the Vietnam Ministry of Health estimated that the resources for more than 80,000 nurses would be insufficient by 2020 [29]. Another study revealed that there are 10–15 patients for every nurse in some large hospitals in Vietnam; in contrast with the standard that each nurse should only care for up to 4 patients to achieve the best care [30]. As a result, nursing is regarded as a particularly stressful occupation in Vietnam, with 87.4% of health workers believing that nursing always requires a high sense of responsibility, heavy work pressure, a much more stressful working environment than other industries, and a dangerous working environment with many potential risk factors for occupational diseases [31]. Furthermore, in a hospital setting, nurses are typically subjected to complex risk factor such as limited JR, including limited job control and conflicts with coworkers, supervisors, and patients, which can lead to adverse health problems [32], and particularly WRMSDs [33]. Recent studies have indicated that WRMSDs have a very high prevalence and incidence among Vietnamese nurses [28,33]. The list of 34 occupational diseases that entitle people to social insurance benefits in Vietnam does still not include MSDs [34].

This study therefore aimed to determine the characteristics of WRMSDs and how JR moderate the association between JD and WRMSD severity among hospital nurses in the Thua Thien Hue province of Vietnam. The intention was to provide sufficient scientific data and evidence to the Vietnam General Confederation of Labor that WRMSDs are a work-related disease and should result in entitlement to social-insurance benefits and other benefits associated with occupational diseases.

## 2. Materials and Methods

### 2.1. Study Design and Sample

This study has a cross-sectional design. Random sampling was used to recruit 265 participants from the target nursing population of 2 hospitals in Thua Thien Hue province of Vietnam. These are the two largest general hospitals in the province of Thua Thien Hue, with about a million patients visiting for examinations and treatment. There are several hospitals in the city, but they are private and lower-class hospitals in districts or villages. As a result, we chose these two hospitals because they are in the same location in city and have the same hospital classification. The sample size was calculated using the following formula: *n* = Z^2^_1−α/2_ × *p* (1 − *p*)/d^2^ = 1.962 × (0.779 × 0.221)/0.052 = 265, where 0.779 comes from the MSD prevalence of 77.9% among nurses in previous studies [28]. Of the 265 questionnaires sent out, 225 were returned (84.9% response rate). The inclusion criteria were nurse working in clinical units/wards as full-time hospital employees, with at least 1 year of experience.

### 2.2. Data Collection

Data were collected by distributing of the self-reporting questionnaires from August to September 2020. The questionnaire included information on MSDs, JD and JR factors, participant demographics, and job characteristics. In this study, PW and psychological demands (PD) were considered JD, and skill discretion, decision authority, supervisor support (SS), and coworker support (CS) were JR. All participants were fully informed about the study’s aims, and the data collection process prior to data collection. Each participant volunteered to participate in the study and was allowed to withdraw at any time. The institutional ethics committee of H university has reviewed and approved this study (H2020/440).

### 2.3. Measurements

#### 2.3.1. Work-Related Musculoskeletal Disorders

The Standardized Nordic Questionnaire (SNQ) was used to determine the presence of musculoskeletal symptoms [35]. The SNQ has been frequently used by health experts to study MSDs in South Africa and elsewhere in the world and is an open, trustworthy, and valid instrument. The questionnaire included items about musculoskeletal issues in nine anatomical regions: neck, shoulders, elbows, wrists/hands, upper back, lower back, hips/thighs, knees, ankles. Participants were asked if they have experienced any WRMSD symptoms (e.g., pain, numbness, tingling, aching, stiffness, or burning) in any of the nine distinct regions during the previous 12 months. If they had, they were further asked about their MSDs in the previous 7 days and if they had prevented them from engaging in regular activities at work and or in leisure time during the previous 12 months.

The reliability of this instrument for this study was evaluated using Cronbach’s alpha, which was 0.93. The author also used a numerical pain rating scale ranging from 0 (no pain) to 10 (severe pain) to determine the intensity of musculoskeletal issues experienced by participants during the previous 12 months.

#### 2.3.2. Job Demands

PW was measured by using the Physical Workload Index Questionnaire [36]. This questionnaire consists of 19 items presented as pictograms. Five of these items are specified in trunk postures, three items in arm positions, five items in leg positions, and six items in weightlifting, with three of the items included the straight-line trunk lifting and three items during 60° trunk lifting. The scoring for this questionnaire was calculated using previously reported formula [34]. All questions are scored on a five-point Likert scale, ranging from never (score of 0) to very often (score of 4). The reliability of the questionnaire was considered satisfactory [36]. This instrument showed good reliability in the present study, with a Cronbach’s alpha of 0.88.

PD were measured with five items on a PD scale derived from the Job Content Questionnaire (JCQ) by Karasek (1985) [35]. The PD scale was described by the workload, intensity, and fragmentation of an excessive workload, hard work, and insufficient working time. Participants were instructed to answer with one of four options for each scale item, ranging from strongly disagree (score of 1) to strongly agree (score of 4). A higher score indicated greater PD severity.

#### 2.3.3. Job Resources

Skill discretion, decision authority, SS, and CS were measured using subscales of the JCQ [37]. Skill discretion comprised six items (learning new things, developing skills, requiring skills, diversity of tasks, repetition, and job creativity), decision authority comprised three items (having freedom to make decisions, choosing how to perform work, and having a lot of say in the job), and SS and CS both included four items. The items were scored on a five-point Likert scale from strongly disagree (score of 1) to strongly agree (score of 4). Mean scores were used, with a higher score indicating higher severity on the respective subscales. Reliability as tested using Cronbach’s alpha yielded values of 0.91, 0.94, and 0.7 for SS, CS, and decision latitude (skill discretion and decision authority), respectively.

#### 2.3.4. Demographics

The job characteristics of participants were collected, including sex, age, marital status, general health status, BMI, working department, working experience (years), job title, job function, and working hours per shift. These were considered confounding variables.

### 2.4. Data Analysis

Descriptive and frequency analyses were used to measure sample characteristics, including individual and job characteristic variables, JD-R factors, and WRMSDs variables. Chi-square was used to compare categorical variables, and Spearman’s correlation coefficients were used to investigate the relationships between JD, JR, and WRMSDs. A simple moderation analysis was conducted in this study using Hayes’ technique [36], in which JR was used as a moderator, with the assumption that it would moderate the path or strength of the association between JD and WRMSD severity regarding pain intensity and the number of anatomical sites of WRMSDs. All continuous variables of predictors and moderators were dichotomized into high and low levels based on median cutoffs, and all confounding variables were included in these analyses. A two-tailed *p* value of 0.05 was considered statistically significant. The moderating impact of the 95% confidence interval (CI) was assessed using 5000 bootstrap samples. SPSS Statistics software (version 20, SPSS, Chicago, IL, USA) and the PROCESS macro (version 3.5.3) of SPSS were used for the data analysis [38].

Moderation analyses as proposed in Hayes’ model 1 were conducted to determine whether JR factors could moderate the relationship between JD and WRMSD severity (Figure 1), while accounting for all covariate variables using the bootstrapping method [38]. Pain intensity and the number of anatomical sites of WRMSDs were considered dependent variables when each of the four JR factors were used to moderate for the relationship between JD and WRMSD severity. Skill discretion, decision authority, SS, and CS were used as the moderators in models 1, 2, 3, 4, respectively.

## 3. Results

Among the 225 participants, 89.3% were female and 88% had normal BMIs (18.5–22.9 kg/m^2^). Their mean age was 35.00 years (SD = 6.39 years), with 52.9% being 30–40 years old. Most participants worked in general wards (internal, surgical, obstetric, or pediatric), while 21.3% worked in either an emergency room, intensive-care unit, or an operating room. Most participants (75.6%) had 5–20 years of working experience. Most nurses (61.3%) worked an average of 8 h each day or shift, with the other 38.7% working more than 8 h per day or shift (Table 1).

JD were measured using PW and PD, which had mean scores of 22.84 (SD = 9.64) and 32.32 (SD = 4.73), respectively. JR was measured using JCQ, with mean scores for skill discretion, decision authority, SS, CS of 34.60 (SD = 2.99), 33.49 (SD = 4.20), 11.87 (SD = 1.64), and 12.56 (SD = 1.27), respectively (Table 1).

Of the 225 participants, 197 (87.6%) had experienced a WRMSD in at least one body region during the previous 12 months. The WRMSD sites with the highest prevalence in these 12 months were the lower back (65.3%), neck (61.8%), shoulder (61.8%), and knee (42.2%) (Figure 2). The prevalence of WRMSDs at a single body site was 11.7%, with two or more sites affected in the other 88.3%. The result also indicated that the mean score of pain intensity during the previous 12 months was 4.11 (SD = 1.75).

The correlation analyses in Table 2 indicate that pain intensity and the number of anatomical sites of WRMSDs were positively correlated with PW-induced JD (*r* = 0.20, *p* < 0.01; *r* = 0.19, *p* < 0.01; respectively), and PD (*r* = 0.32, *p* < 0.01; *r* = 0.16, *p* < 0.05; respectively). Not all JR variables were significantly correlated with pain intensity (*p* > 0.05). Only CS was correlated with the number of anatomical sites of WRMSDs (*r* = 0.16, *p* < 0.05) (Table 2).

Our results indicated that only model 4 had a significant main effect of PW on pain intensity and the number of anatomical sites of MSDs (*p* < 0.05, *p* < 0.05, respectively), and also a significant interaction between CS and PW on pain intensity and the number anatomical sites of WRMSDs (*p* = 0.02, *p* < 0.01, respectively) (Table 3). The conditional effects of PW, alongside the moderator of CS on WRMSD severity, are also listed in Table 3. For pain intensity, our results indicated that 95% CI for the difference between the moderating effects did not cross zero when CS was low. This suggests that pain intensity significantly increased when PW was high and CS was low. In contrast, when CS was high, this moderating effect was not significant (β = −0.61, 95% CI =−1.87 to 0.66). Similarly, for the number of anatomical sites of WRMSDs, when CS was low, the 95% CI for the difference between the moderating effects did not cross zero, whereas it did when the CS moderator was high. This indicated that the number of anatomical sites of WRMSDs significantly increased when PW was high and CS was low. In contrast, when CS was high, the effect was not significant (β = −1.08, 95% CI = −2.46 to 0.30). Plots of these interactions are shown in Figure 3.

The result of this analysis indicated that when PD were used as an independent variable, most PD were independently related to the increasing severity of pain in models 2, 3, and 4. However, JR did not significantly reduce WRMSD severity. The buffering roles of JR in the relationship between PD and WRMSD severity were not significant in all models (Table 4).

## 4. Discussion

The WRMSDs of nurses were measured in this study at nine locations: neck, shoulders, upper back, elbows, wrists/hands, lower back, hips, knees, and ankles. WRMSDs were common among the included hospital nurses, with 87.6% experiencing it in at least one region during the previous 12 months, most commonly in the lower back (65.3%), neck and shoulder (61.8%), and knee (42.2%). This prevalence was consistent with an earlier report of WRMSDs and lower back pain occurring in 88% and 65.3% of nurses in in Sweden, respectively [39]. Attar (2014) suggested that WRMSDs had a prevalence of 85.0%, with the lower-back symptoms being the most common among Saudi Arabian nurses (65.7%) [40]. The prevalence of WRMSDs during the previous 12 months among hospital nurses in our research was also comparable with the findings of other studies in Asian countries [41,42,43]. Its prevalence was higher in Malaysian nurses (73.1%), with common sites in the neck (48.9%), upper back (40.7%), and shoulder (36.9%) [44]. In Vietnam, where 74.7% of participants reported MSDs during the previous 12 months, the most-common sites were the lower back (44.4%), neck (44.1%), upper back (32.7%), and shoulder (29.7%) [28]. These differences are attributable to variation in the characteristics of hospitals, diverse working conditions, nurse-care activity demands across countries, and cultural views of MSD symptoms.

The present study found that PW and PD were positively correlated with WRMSD severity. This suggests that nurses who work in environments with high JD have greater WRMSD severity. This is consistent with Hsien’s (2016) finding that the severity and symptoms of MSDs were significantly correlated with workload intensity among nurses from a hospital in Taoyuan [45]. Cantley et al. (2016) found that the likelihood of major injuries and severe MSDs requiring medical treatment, job limitations, or less working time was 49% higher in cases with high PD [46]. Another study involving Iranian hospital nurses at the Semnan University of Medical Sciences indicated that the more nurses experienced psychological burden and role-related pressures, the more MSD complaints they experienced [26]. The results from the present study imply that intervention programs to alleviate WRMSD severity among nurses should include strategies to reduce both physical risk factors and PD.

Skill discretion, decision authority, and SS did not appear to be significantly correlated with WRMSD severity in our study, which contrasts with many previous studies indicating that these factors are associated with WRMSDs. Sherehiy et al. (2004) found that psychosocial factors such as work organization issues, social relations at work, and job control were associated with musculoskeletal problems [47]. Parkes (2008) suggested that SS was a strong negative predictor, that higher SS levels were associated with lower MSD occurrence, and that higher job control was related to a lower risk of MSDs [48]. Cantley et al. (2016) reported the risks of injury and MSDs were higher in those with poor job control. This disparity can be explained by the situation of Vietnamese nurses, where the importance of psychosocial factors has not been recognized or promoted [49]. For nurses with high levels of experience in Vietnam’s healthcare system, the perception of low job control and low social support are sensitive issues. They rarely have a voice regarding their role and position in the system, which has gradually become a problem that they have to accept. The position and role of nurses within healthcare system has improved recently due to improvements in nursing education and training, but asserting their self-determining role within the healthcare system still requires time. This is consistent with the current study finding that WRMSDs are often related to physical work or work pressure rather than social support and job control.

Our study hypothesis was that JR can moderate the associations between JD and WRMSDs while adjusting for confounding factors. The results indicate that CS could moderate the association between PW and WRMSD severity (*p* < 0.05), with WRMSD severity being significantly higher when PW interacted with low CS. These results mean that if nurses feel a lack of support from their work colleagues, they are less likely to consider these relationships when carrying out their nursing activities, and they therefore carry out their work independently even in difficult situations, which leads to high workload perceptions and reduced energy and exhaustion. Feeney and Collins (2015) found that CS can help employees to improve and enhance their capacity to cope effectively even in difficult circumstances [50]. This finding was also congruent with Pekkarinen’s finding that high PW was associated with an increased risk of musculoskeletal complaints among nurses when social support was too poor. Coworkers play an important role in performing daily physical tasks such as carrying, moving, or transferring patients [51]. However, the current study indicated that no variable moderated the correlation between PD and WRMSD severity, despite PD being an independent predictor of pain intensity.

Despite the lack of evidence on the buffering effect of CS on the relationship between JD and WRMSDs, many previous studies have suggested that CS moderates the effects of JD on psychological health, such as stress or burnout. According to Yang et al. (2015), in order to cope with workplace pressures, employees collaborate with coworkers to reduce work-related complaints and presenteeism; they perceive that they are supported by their colleagues and thus flourish in their work and do better than those who receive less support [52]. Blomberg and Rosander (2020) similarly indicated that the relationship between exposure to bullying and health and well-being may be moderated if the perceived support received from working colleagues is low, which could affect the negative impacts of bullying on general health and well-being [53]. These results highlight the role of CS as an important psychosocial resource for alleviating the negative effects of JD on the health of nurses, including their mental and physical health. This may include providing adequate CS to nurses, especially when JD cannot feasibly be reduced. This is also consistent with the JD-R model, in which interactions between JD and psychosocial factors affect health outcomes [54]. This implies that PW demands lead to health problems due to the increasing efforts required to fulfill responsibilities. Moreover, JR provide employees with solutions to meet demands and safeguard them from these health problems [20].

In conclusion, the findings of this study emphasize the role of CS as a buffer against the effects of PW on WRMSD severity, which is consistent with the assumption of the JD-R model [22,54,55]. Furthermore, the present study has added to the literature by examining the influence of JD and JR on the mental and physical health of employees, thereby filling research gaps in the JD-R model. An intervention program for nurses to minimize WRMSD severity should therefore incorporate efforts to lower both physical risk factors and psychological pressures. Nurses should also be provided with adequate CS to handle their increasing PW, and to reduce WRMSD severity. Furthermore, managers should focus on other ergonomic risk factors such as providing a supportive and safe working environment [56].

This study was subject to several limitations. First, a cross-sectional design is well known to be incapable of establishing a causal relationship between risk factors and WRMSDs. Longitudinal designs are therefore recommended for future studies to confirm the current findings. Second, data were collected using self-reporting measures. The responses might have been misleading due to respondents providing more acceptable answers rather than factual responses about their actual experiences. Additionally, it is not possible to completely rule out that MSDs are caused by factors outside of work. Establishing an intervention to prevent WRMSD occurrence among Vietnamese nurses was inadequate. Future research needs to focus on methods for preventing WRMSDs among the nursing population. Furthermore, our study did not look at the effects of musculoskeletal problems on nurses’ health and nursing care efficacy. Therefore, future research needs to further investigate the relationship between musculoskeletal disorders and nursing care performance.

On the contrary, this study has some strengths. This is one of the first studies conducted in nursing on musculoskeletal disorders in Vietnam. This is the foundation for larger-scale investigations to develop appropriate and successful intervention strategies. The study also provided meaningful information to propose musculoskeletal disorders on the list of occupational diseases that need to be entitled to benefits from social insurance.

## 5. Conclusions

This study aimed to determine the moderating effects of job resources (JR) in the relationship between job demands (JD) and the severity of Work-Related Musculoskeletal Disorders (WRMSDs) among hospital nurses in the Thua Thien Hue province, Vietnam. The simple moderation analysis highlighted the moderating role of CS on the relationship between PW and WRMSD severity, demonstrating that WRMSDs severity significantly increased when PW was high alongside a low level of CS. The current study indicated that nurses should be provided with appropriate CS to minimize WRMSDs severity. Furthermore, an intervention program for managing WRMSDs in nurses may involve minimizing physical risk factors and PDs.

## Figures and Tables

**Figure 1 ijerph-19-04774-f001:**
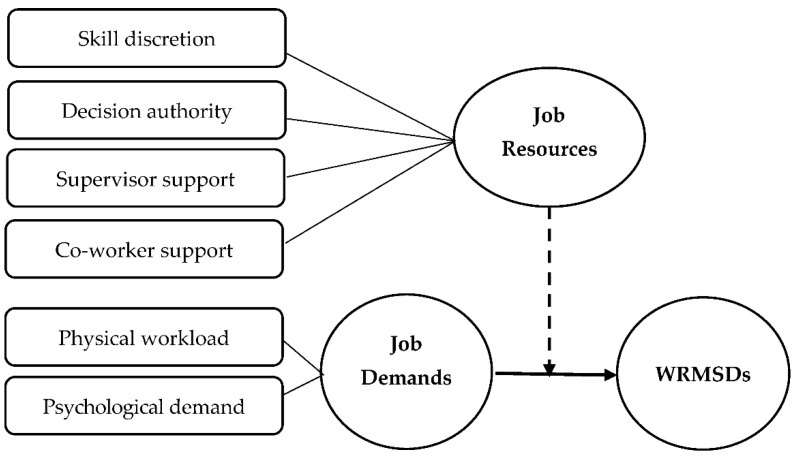
The proposed moderated mediation model.

**Figure 2 ijerph-19-04774-f002:**
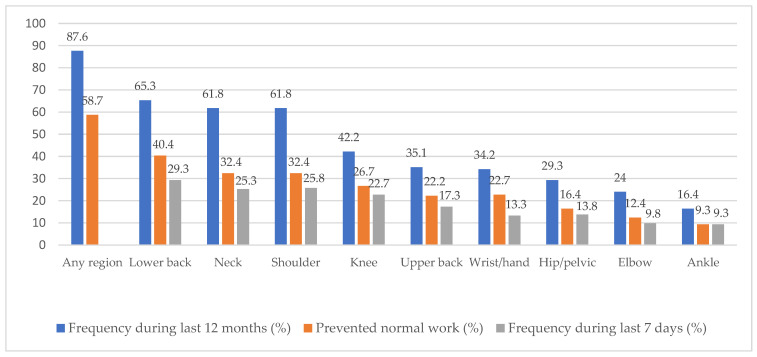
Body region and WRMSDs among study participants (*N* = 225).

**Figure 3 ijerph-19-04774-f003:**
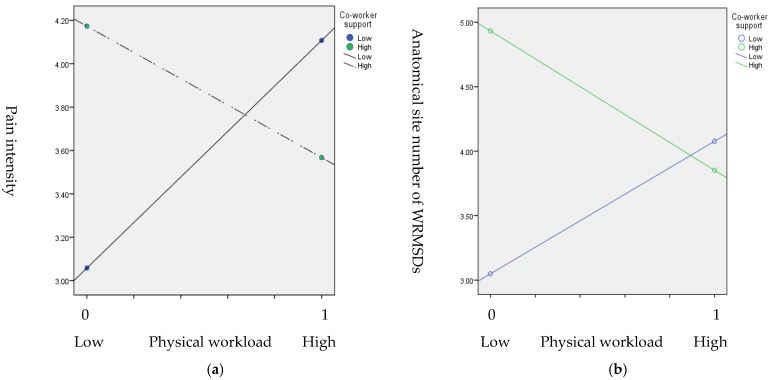
Interaction effect between PW and CS on WRMSD severity: (**a**) the pain intensity; (**b**) the number of anatomical sited of WRMSDs.

**Table 1 ijerph-19-04774-t001:** Descriptive statistics of studied variables of study participants (*N* = 225).

Variables	Category	*N* (%)	Mean (SD)
Gender	Male	24 (10.7)	
	Female	201 (89.3)	
Age	<30 age	61 (27.1)	
	30–40 age	119 (52.9)	
	>40 age	45 (20.0)	
BMI	Underweight (<18.5)	15 (6.7)	
	Normal (18.5–22.9)	172 (76.4)	
	Overweight and obese (>23)	38 (16.9)	
Working department	General ward	135 (60.0)	
	ER-ICU-OP *	48 (21.3)	
	Other	42 (18.7)	
Working experience	<5 years	37 (16.4)	
	5–20 years	170 (75.6)	
	>20	18 (8.0)	
Working hour per shift	<8 h	138 (61.3)	
	>8–12 h	62 (27.6)	
	>12 h	25 (11.1)	
Job demands	Physical workload [0–56.17]		22.84 (9.64)
	Psychological demand [12–48]		32.32 (4.73)
Job resources	Skill discretion [12–48]		34.60 (2.99)
	Decision authority [12–48]		33.49 (4.20)
	Supervisor support [4–16]		11.87 (1.64)
	Coworker support [4–16]		12.56 (1.27)
Severity of WRMSDs	Pain intensity		4.11 (1.75)
	Anatomical site number		4.23 (2.38)

[ ] indicates the full possible range of the scale score. * ER, Emergency room; ICU, intensive care unit; OR, operating room.

**Table 2 ijerph-19-04774-t002:** Results of the correlation analyses of job demands, job resources and the WRMSD severity among study participants (*N* = 225).

Variables	Mean	SD	WRMSD Severity (DVs)			
			Pain Intensity		Anatomical Sites Number	
			*r*	*p*	*r*	*p*
Job demand (IVs)						
1. Physical workload	22.84	9.6	0.20	<0.01	0.19	<0.01
2. Psychological demand	32.32	4.7	0.32	<0.01	0.16	<0.05
Job resources (W)						
3. Skill discretion	34	2.99	0.13	>0.05	0.08	>0.05
4. Decision authority	33	4.20	−0.01	>0.05	0.05	>0.05
5. Supervisor support	11.87	1.64	−0.06	>0.05	−0.01	>0.05
6. Coworker support	12.56	1.30	0.08	>0.05	0.16	<0.05

IVs—independent variables; DVs—dependent variables; W—moderator.

**Table 3 ijerph-19-04774-t003:** Moderating analysis of each JR variable between PW and WRMSD severity (*N* = 225).

Variables	WRMSD Severity			
	Pain Intensity		Anatomical Site Number	
Model ^#^	B	SE	B	SE
Moderating model 1	R^2^ = 0.16, F_(10,_ _214)_ = 2.80 ***		R^2^ = 0.15, F_(10, 214)_ = 2.64 ***	
Physical workload (PW)	0.44	0.40	0.77	0.48
Skill Discretion (SKD)	0.33	0.50	0.68	0.48
PW × SKD	0.29	0.63	−0.70	0.72
Moderating model 2	R^2^ = 0.15, F_(10, 214)_ = 2.70 ***		R^2^ = 0.16, F_(10, 214)_ = 2.72 ***	
Physical workload (PW)	0.37	0.37	0.25	0.43
Decision Authority (DA)	−0.15	0.42	0.21	0.47
PW × DA	0.61	0.59	0.69	0.70
Moderating model 3	R^2^ = 0.15, F_(10, 214)_ = 2.73 ***		R^2^ = 0.16, F_(10, 214)_ = 2.86 ***	
Physical workload (PW)	0.49	0.34	0.62	0.42
Supervisor support (SS)	−0.19	0.53	0.99	0.69
PW × SS	0.71	0.74	−0.45	0.94
Moderating model 4	R^2^ = 0.17, F_(10,_ _214)_ = 3.13 ***		R^2^ = 0.14, F_(10, 214)_ = 4.81 ***	
Physical workload (PW)	1.05 *	0.35	1.49 *	0.41
Coworker support (CS)	1.12 *	0.60	1.03 **	0.59
PW × CS	−1.65 *	0.72	−2.11 **	0.78
Conditional effects of the PW at values of the moderator of CS on WRMSD severity				
Coworker support	Effect (Boot SE)		Boot 95% CI	
Low	1.05 (0.35)	0.37; 1.73	1.03 (0.41)	0.21; 1.84
High	−0.61 (0.64)	−1.87; 0.66	−1.08 (0.70)	−2.46; 0.30

Note: * *p* < 0.05; ** *p* < 0.01; *** *p* < 0.001; B = regression coefficient; SE = standard error of regression coefficient; bootstrap sample size = 5000; Model ^#^: These models were analyzed after controlling all covariate variables.

**Table 4 ijerph-19-04774-t004:** Moderating analysis of each JR variable between PD and WRMSD severity (*N* = 225).

Variables	WRMSD Severity			
	Pain Intensity		Anatomical Site Number	
Model ^#^	B	SE	B	SE
Moderating model 1	R^2^ = 0.17, F_(18,_ _206)_ = 2.89 ***		R^2^ = 0.14, F_(18,_ _206)_ = 2.33 **	
Psychological demand (PD)	0.72	0.39	0.21	0.50
Skill discretion (SD)	0.14	0.43	0.58	0.58
PD × SD	0.30	0.61	−0.43	0.78
Moderating model 2	R^2^ = 0.17, F_(18,_ _206)_ = 2.92 ***		R^2^ = 0.16, F_(18,_ _206)_ = 2.55 ***	
Psychological demand (PD)	1.20 **	0.36	0.62	0.79
Decision authority (DA)	0.40	0.39	0.28	0.53
DL × DA	−0.64	0.56	−0.56	0.86
Moderating model 3	R^2^ = 0.10, F_(18,_ _206)_ = 3.03 ***		R^2^ = 0.15, F_(18,_ _206)_ = 2.64 ***	
Psychological demand (PD)	0.91 **	0.32	0.27	0.39
Supervisor support (SS)	0.13	0.53	1.12	0.71
PD × SS	0.10	0.71	−0.74	0.85
Moderating model 4	R^2^ = 0.17, F_(18,_ _206)_ = 2.84 ***		R^2^ = 0.15, F_(18,_ _206)_ = 2.57 ***	
Psychological demand (PD)	0.75 *	0.33	0.24	0.42
Coworker support (CS)	−0.22	0.47	0.99	0.63
PD × CS	0.63	0.65	−0.57	0.82

Note: * *p* < 0.05; ** *p* < 0.01; *** *p* < 0.001; B—regression coefficient; SE—standard error of regression coefficient; Model ^#^: These models were analyzed after controlling all covariate variables.

## Data Availability

The data that support the findings of this study are available from the corresponding author upon reasonable request.

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
