# Peer review of "The Role of Job Resources in the Relationship between Job Demands and Work-Related Musculoskeletal Disorders among Hospital Nurses in Thua Thien Hue Province, Vietnam"

_ijerph, 2022, doi:10.3390/ijerph19084774_

Round 1

Reviewer 1 Report

  The manuscript submitted for review provides a valuable discussion of the exposure of nurses to work-related musculoskeletal disorders. In the introductory part, the authors thoroughly and comprehensively discussed the basics of the issue, as well as the essence of the models constituting the basis of the conducted research and the general purpose of the work. The second part of the work contains a description of the research methodology. The study design was very well presented by the authors. I would only consider adding information about the characteristics of hospitals in which respondents were recruited (first of all, are we dealing with general hospitals?). I also did not find information about the time in which the study was carried out. Moreover, I would like to ask the authors to explain the meaning of the symbol 'Z21', which was included in the formula on the basis of which the size of the tested sample was determined. The results of the study were presented in a clear and understandable way. They also correspond with the conclusions drawn by the authors. The discussion referred to numerous sources, showing both the similarities and differences to the results obtained by other researchers. The observed differences were explained using convincing arguments. In conclusion, I would like to emphasize that, apart from the small remarks I have quoted above, I consider the reviewed article to be a very good study, both in terms of content and form.

Author Response

Responses to Reviewers

Title: The role of coworker support in the relationship between physical workload and work-related musculoskeletal disorders among hospital nurses in Vietnam

Thank you for sending us the comments of the reviewers. Our responses to each comment are provided below. Changes made in the text are shown with track change, except for changes made solely to improve the use of English.

Reviewer #1

  1. I would only consider adding information about the characteristics of hospitals in which respondents were recruited (first of all, are we dealing with general hospitals?).

>> On line 105 on page 3, the information of choosing two hospitals in Thua Thien Hue Province was explained.

  1. I also did not find information about the time in which the study was carried out.

>> The study was conducted from August to September 2020 on the line 116 page 3.

  1. Moreover, I would like to ask the authors to explain the meaning of the symbol 'Z21', which was included in the formula on the basis of which the size of the tested sample was determined.

>> The formula was corrected on line 110 on page 3.

Reviewer 2 Report

Author should be the reason of the 2 hospitals selection in only one province in this study.

Author should be added the ethical consideration in the methodology.

Author should be rechecked the reference citation in whole manuscript.

Author should be added the recommendation from your result finding.

Author Response

Responses to Reviewers

Title: The role of coworker support in the relationship between physical workload and work-related musculoskeletal disorders among hospital nurses in Vietnam

Thank you for sending us the comments of the reviewers. Our responses to each comment are provided below. Changes made in the text are shown with track change, except for changes made solely to improve the use of English.

Reviewer #2

  1. Author should be the reason of the 2 hospitals selection in only one province in this study.

>>On line 105 on page 3, the information of choosing two hospitals in Thua Thien Hue Province was explained. The reason why we only select two hospitals inn Thua Thien Hue province. There are several hospitals in the city, but they are private and lower-class hospitals in districts or villages. As a result, we chose these two hospitals because they are in the same location in city and have the same hospital classification.

  1. Author should be added the ethical consideration in the methodology.

>> Comment accepted. On line 120 on page 3, we added information of ethical consideration in our study.

  1. Author should be rechecked the reference citation in whole manuscript.

>>Done. The references was re-checked and corrected for whole manuscript.

  1. Author should be added the recommendation from your result finding.

>> On line 365 and 379 on page 13, there are some suggestions from our study finding. That an intervention program for nurses to minimize WRMSD severity should therefore in-corporate efforts to lower both physical risk factors and psychological pressures. Nurses should also be provided with adequate CS to handle their increasing PW, and to reduce WRMSD severity. Furthermore, managers should focus on other ergonomic risk factors such as providing supportive and safe working environment.

Reviewer 3 Report

It is obvious that nurses’ mental and physical health problems with musculoskeletal diseases and vocational stressors are key factors in reducing the quantity and quality of their work performance, especially patient care. Therefore, the prevalence of MSDs in nurses and their association with some personal and professional factors need to be studied to prevent and treat them in this sensitive group. In this study, it is indicated in the Introduction that the authors are collecting evidence on this health hazard to be part of the occupational diseases that entitle people to social insurance benefits in Vietnam.

Title: The role of coworker support in the relationship between physical workload and work-related musculoskeletal disorders among hospital nurses in Vietnam

Comment: This report is well written but somehow lack of coherence between aim and title

Background: This cross-sectional study aimed to determine the effects of job resources (JR) and job demands (JD) on the severity of Work-related musculoskeletal disorders (WRMSDs) among hospital nurses in Vietnam. (2) Methods: Data were collected via a self-reporting questionnaire administrated to 225 nurses from 2 hospitals in the Thua Thien Hue province of Vietnam from August to September 2020. The questionnaire included infor-mation on musculoskeletal symptoms, JD and JR, and the demographics and job characteristics of participants. The simple moderation analysis was used for data analysis. (3) Results: The findings indicated that 87.6% of nurses had experienced WRMSDs in at least one body region during the previous 12 months, with lower back (65.3%), neck (61.8%), and shoulders (61.8%) being the three most-common sites. physical workloads (PW) and psychological demands (PDs) were significantly correlated with WRMSDs severity (p < 0.05). This study further underlined the moderating role of coworker support (CS) on the relationship between PW and WRMSD severity. WRMSDs severity significantly increased when PW was high alongside a low level of CS. (4) Conclusions: The current study indicated that nurses should be provided with appropriate CS to minimize WRMSDs severity. Furthermore, an intervention program for managing WRMSDs in nurses may involve minimizing physical risk factors and PDs.

Comment: 2 hospitals in the Thua Thien Hue province of Vietnam : why these hospitals were chosen? Conclusion not in congruent with the aim

 Previous studies have indicated that the occurrence of WRMSDs is frequently com-39 plex and affected by individual characteristics and physical and psychosocial risk factors 40 [12, 13].

Comment: suggest to use this SRMA as reference: Relationship Between the Exposure to Occupation-related Psychosocial and Physical Exertion and Upper Body Musculoskeletal Diseases in Hospital Nurses: A Systematic Review and Meta-analysis,

Asian Nursing Research, Volume 15, Issue 3, https://www.sciencedirect.com/science/article/pii/S1976131721000220

This study was subject to several limitations. First, a cross-sectional design is well-known to be incapable of establishing a causal relationship between risk factors and WRMSDs. Longitudinal designs are therefore recommended for future studies to confirm the current findings. Second, data were collected using self-reporting measures. The responses might have been misleading due to respondents providing more-acceptable answers rather than factual responses about their actual experiences. Besides, it is not possible to completely rule out that MSDs are caused by factors outside of work. Establishing an intervention to prevent WRMSD occurrence among Vietnamese nurses was inade-346 quate. Future, research needs to focus on methods for preventing WRMSDs among the nursing population.

Comment: The weaknesses of the study were exposed in such a way that does not allow us to consider it for publication. Please add solution/strength of the study too. Suggest to refer to

Yang, M. H., Jhan, C. J., Hsieh, P. C., & Kao, C. C. (2021). A Study on the Correlations between Musculoskeletal Disorders and Work-Related Psychosocial Factors among Nursing Aides in Long-Term Care Facilities. International journal of environmental research and public health19(1), 255. https://doi.org/10.3390/ijerph19010255

This study has a cross-sectional design. Random sampling was used to recruit 265  participants from the target nursing population of 2 hospitals in Thua Thien Hue province 101 of Vietnam. The sample size was calculated using the following formula: n= Z21 - α/2 × p 102 (1 – p)/d2 = 1.962 × (0.779 × 0.221)/0.052 = 265, where 0.779 comes from the MSD prevalence 103 of 77.9% among nurses in previous studies [27]. Of the 265 questionnaires sent out, 225 104 were returned (84.9% response rate). The inclusion criteria were nurse working in clinical 105 units/ wards as full-time hospital employees, with at least 1 year of experience.

Comments: how to justify the sample calculation if the response rate is not considered in the calculation?  

Statistical Analyses: in the existing study, bootstrap was used. Did it really required? Please refer to the following stat method:

SPSS for Windows version 22.0 (IBM Corp., Armonk, NY, USA) was used for data analysis, including the calculation of descriptive statistics, independent t test, one-way ANOVAs, and the calculation of Pearson’s correlation coefficients. Furthermore, hierarchical regression analysis was conducted to identify the amount of variation in MSDs among nursing aides in LTCFs. In the hierarchical regression analysis, the dependent variable was the MSD score, and the independent variables were demographic characteristics and the job background of nursing aides, which have reached significance variables, and the work-related psychological factors. The Kolmogorov–Smirnov normality test, Quantile–Quantile plot, Durbin–Watson test, and Residual Plot test were used to check whether the residual errors of regression were normally distributed, independent and homoscedastic. According to the collinearity diagnostics, tolerance results > 0.1 and variance inflation factor (VIF) results < 10, no collinearity was associated with the three models used in this study.

Figure 1. The proposed moderated mediation model: this is overly too simplified

Figure 2. Body region and WRMSDs among study participants (N=225) -arrange from the largest to lowest

Table 1. Descriptive statistics of studied variables of study participants (N=225)

Table 2. Inter-correlations, means, and standard deviations among study variables

Table 4. Moderating analysis of each JR variable between PD and WRMSD severity (N=225)

The tables above should be trimmed

 Conclusions

This study aimed to determine the effects of job resources (JR) and job demands (JD) 350 on the severity of Work-related musculoskeletal disorders (WRMSDs) among hospital 351 nurses in Vietnam. From the simple moderation analysis, the moderating role of CS on 352 the relationship between PW and WRMSD severity. WRMSDs severity significantly in-353 creased when PW was high alongside a low level of CS. The current study indicated that 354 nurses should be provided with appropriate CS to minimize WRMSDs severity. Further-355 more, an intervention program for managing WRMSDs in nurses may involve minimizing 356 physical risk factors and PDs.

Comment: please restructure the conclusion to reflect the main gist of the findings.

Author Response

Responses to Reviewers

Title: The role of coworker support in the relationship between physical workload and work-related musculoskeletal disorders among hospital nurses in Vietnam

Thank you for sending us the comments of the reviewers. Our responses to each comment are provided below. Changes made in the text are shown with track change, except for changes made solely to improve the use of English.

Reviewer #3

  1. Title: The role of coworker support in the relationship between physical workload and work-related musculoskeletal disorders among hospital nurses in Vietnam. This report is well written but somehow lack of coherence between aim and title

>> The title and aim of the current study were modified in which that reflex each other. At page 1/14

  1. hospitals in the Thua Thien Hue province of Vietnam : why these hospitals were chosen? Conclusion not in congruent with the aim
  2. Previous studies have indicated that the occurrence of WRMSDs is frequently com-39 plex and affected by individual characteristics and physical and psychosocial risk factors 40 [12, 13].

suggest to use this SRMA as reference: Relationship Between the Exposure to Occupation-related Psychosocial and Physical Exertion and Upper Body Musculoskeletal Diseases in Hospital Nurses: A Systematic Review and Meta-analysis,

Asian Nursing Research, Volume 14, Issue 3, https://www.sciencedirect.com/science/article/pii/S1976131721000220

>> On line 43 on page 1, a new reference was added.

  1. This study was subject to several limitations. First, a cross-sectional design is well-known to be incapable of establishing a causal relationship between risk factors and WRMSDs. Longitudinal designs are therefore recommended for future studies to confirm the current findings. Second, data were collected using self-reporting measures. The responses might have been misleading due to respondents providing more-acceptable answers rather than factual responses about their actual experiences. Besides, it is not possible to completely rule out that MSDs are caused by factors outside of work. Establishing an intervention to prevent WRMSD occurrence among Vietnamese nurses was inade-346 quate. Future, research needs to focus on methods for preventing WRMSDs among the nursing population.

The weaknesses of the study were exposed in such a way that does not allow us to consider it for publication. Please add solution/strength of the study too. Suggest to refer to

Yang, M. H., Jhan, C. J., Hsieh, P. C., & Kao, C. C. (2021). A Study on the Correlations between Musculoskeletal Disorders and Work-Related Psychosocial Factors among Nursing Aides in Long-Term Care Facilities. International journal of environmental research and public health19(1), 255. https://doi.org/10.3390/ijerph19010255

>> On line 369 to 387 on 13, we revised the limitation and strengths of the study.

  1. This study has a cross-sectional design. Random sampling was used to recruit 265 participants from the target nursing population of 2 hospitals in Thua Thien Hue province of Vietnam. The sample size was calculated using the following formula: n= Z21 - α/2 × p 102 (1 – p)/d2 = 1.962 × (0.779 × 0.221)/0.052 = 265, where 0.779 comes from the MSD prevalence of 77.9% among nurses in previous studies [27]. Of the 265 questionnaires sent out, 225 were returned (84.9% response rate). The inclusion criteria were nurse working in clinical units/ wards as full-time hospital employees, with at least 1 year of experience.

Comments: how to justify the sample calculation if the response rate is not considered in the calculation?  

>>On line 112 on page 3, we added some information of explaination about same calculation that we intend to distribute the questionnaires directly to the study participants and collect them immediately while determining the sample size and sampling procedure. Therefore the non-response rate is 0%. However, in fact 14.1% of participants did not return the questionaire due to some reasons and/or provided inadequate information. 

  1. Statistical Analyses: in the existing study, bootstrap was used. Did it really required? Please refer to the following stat method:

>> On line 188 on page 4, we used moderation analysis as suggested by Hayes (2018) to examine the moderating role of JR on the relationship between JD and WRMSDs. That is why we explain this method.

  1. Figure 1. The proposed moderated mediation model: this is overly too simplified

>> The figure was modified at page: 5.

  1. Figure 2. Body region and WRMSDs among study participants (N=225) -arrange from the largest to lowest

Author’s response: The figure was modified at page: 7.

  1. Table 1. Descriptive statistics of studied variables of study participants (N=225)

Table 2. Inter-correlations, means, and standard deviations among study variables

Table 4. Moderating analysis of each JR variable between PD and WRMSD severity (N=225)

The tables above should be trimmed

>> We trimmed the table as reviewer’s comments. At page 6, 8.

  1. Conclusions: please restructure the conclusion to reflect the main gist of the findings.

>> The conclusion part was revised to reflect the main findings. At page 13.
